# Long-Term Indoor-Outdoor PM_2.5_ Measurements Using PurpleAir Sensors: An Improved Method of Calculating Indoor-Generated and Outdoor-Infiltrated Contributions to Potential Indoor Exposure

**DOI:** 10.3390/s23031160

**Published:** 2023-01-19

**Authors:** Lance Wallace, Wayne Ott

**Affiliations:** 1Independent Researcher, 428 Woodley Way, Santa Rosa, CA 95409, USA; 2Department of Civil and Environmental Engineering, Stanford University, 1008 Cardiff Lane, Redwood City, CA 94061, USA

**Keywords:** exposure, indoor particles, infiltration factor, PM_2.5_, PurpleAir, Random Component Superposition (RCS), Plantower sensors, indoor-generated particles

## Abstract

Low-cost monitors make it possible now for the first time to collect long-term (months to years) measurements of potential indoor exposure to fine particles. Indoor exposure is due to two sources: particles infiltrating from outdoors and those generated by indoor activities. Calculating the relative contribution of each source requires identifying an infiltration factor. We develop a method of identifying periods when the infiltration factor is not constant and searching for periods when it is relatively constant. From an initial regression of indoor on outdoor particle concentrations, a Forbidden Zone can be defined with an upper boundary below which no observations should appear. If many observations appear in the Forbidden Zone, they falsify the assumption of a single constant infiltration factor. This is a useful quality assurance feature, since investigators may then search for subsets of the data in which few observations appear in the Forbidden Zone. The usefulness of this approach is illustrated using examples drawn from the PurpleAir network of optical particle monitors. An improved algorithm is applied with reduced bias, improved precision, and a lower limit of detection than either of the two proprietary algorithms offered by the manufacturer of the sensors used in PurpleAir monitors.

## 1. Introduction

Airborne fine particles are the single most important environmental cause of mortality. Outdoor and indoor sources are roughly equally important, with annual estimated deaths in the millions [1,2]. The London smog of the 1950s, which killed thousands of persons over a period of a week, was instrumental in activating many countries to monitor outdoor particle concentrations [3]. Indoor risks were perhaps slower to be recognized and still receive only a fraction of the attention and resources devoted to monitoring and controlling sources of outdoor particle pollution. However, most people spend most of their lives indoors [4]. Therefore, multiple studies have been carried out to measure either personal or indoor exposures to particles [5,6,7,8,9,10,11,12]. However, these studies have been limited to short-term measurements (days or weeks), since the expense of maintaining research-grade instrumentation in homes, together with the noise and clutter of the instruments, has made long-term (months or years) studies with actual measurements of indoor concentrations nearly impossible.

Recently, however, the advent of low-cost monitors has made it possible to measure PM_2.5_ continuously for months or even years inside and outside residences. Multiple manufacturers have produced optical particle counters employing tiny noiseless sensors to estimate fine particle concentrations. These monitors have been tested for response to particles produced by multiple common indoor sources and many have performed well [13,14]. Their unobtrusive size and lack of noise has made them attractive to homeowners interested in their exposure to fine particles.

An important goal in studies of human exposure to particles has been to differentiate particles produced by indoor sources from those that infiltrate homes from outdoors. Multiple methods have been adopted. In the pioneering Harvard 6-City Study, indoor air concentrations were regressed on outdoor concentrations [15]. The slope of the regression was interpreted as an infiltration factor *F_inf_* that could be multiplied by the outdoor concentration to determine the contribution of outdoor particles to total indoor concentration.

Alternative methods have also been applied, such as using a tracer (typically sulfur) with few indoor sources to reduce the indoor-generated particle concentrations [16,17,18,19]; using a “censoring” method on continuous indoor measurements to eliminate indoor sources [20,21,22,23]; using a recursive modeling approach based on dynamic modeling of near-continuous measurements [24,25]; measuring deposition and penetration directly [26]; applying mass balance analysis to estimate resuspension [27,28,29]; and using LOESS regression to detect minima in the indoor/outdoor ratio [30]. 

However, the regression method is still being widely used. A careful statistical analysis of the regression method, called the Random Component Superposition (RCS) method is provided in [31]. In this approach, the following model is used:*C_in_*(*t*) = *Pa*/(*a* + *k*) *C_out_*(*t*) + *G*(*t*) (1)
where *C_in_* and *C_out_* are the instantaneous indoor and outdoor concentrations (µg/m^3^), *P* is the penetration factor (dimensionless number describing the fraction of outdoor particles able to cross the building envelope), *a* is the air exchange rate (h^−1^), *k* is the deposition rate (h^−1^), and *G* is the instantaneous concentration at time *t* caused by indoor activities. 

The term *Pa/(a + k)* is the infiltration factor *F_inf_* and describes the fraction of outdoor PM that enters the home. In the case of a time-averaged monitoring period, the equation may be written
<*C_in_*> = *F_inf_* <*C_out_*> + <*G*>(2)
where now <*C_in_*> and <*C_out_*> are the averaged values, <*G*> is the average contribution of indoor activities to the indoor concentration, and *F_inf_* is considered to be constant. This equation is in the form of a simple linear regression, where the intercept is the average contribution of indoor-generated particles to the total indoor concentration, and the slope is the infiltration factor *F_inf_*. Multiplying the observed outdoor average concentration by *F_inf_* determines the indoor average concentration of infiltrated particles, or *C_inf_*:<*C_inf_*>= *F_inf_* <*C_out_*> (3)

Health effects depend on exposure. Remembering that exposure is defined as the contact of a pollutant with a receptor (in this case fine particles in the breathing zone of a human), for a person over time *T*, a rough estimate of exposure would be given by
<*C_in_*>*T* = *F_inf_* <*C_out_*>*T* + <*G*>*T*
(4)

Of course, people are not always indoors, so Equation (4) is not a measure of their exposure. It is, however, a measure of the *potential* exposure of a person indoors over time.

In particular, the first term on the right-hand side of Equation (4) is described by epidemiologists as the exposure to particles of ambient origin. Epidemiologists then seek a relationship between health effects and this exposure to particles of ambient origin. The second term is the potential exposure to indoor-generated particles. Epidemiologists do not generally include this second term, since *G* is usually completely unknown.

There are three sources of error in this formulation. First, indoor-generated concentrations are seldom known. These can sometimes be substantial, as in smoking tobacco or marijuana, cooking on gas or electric stoves, burning candles, or simply moving about and resuspending particles [32,33,34,35,36,37,38]. Secondly, little is known about the times the residents are not at home. Thirdly, the equation assumes a constant infiltration factor, despite a number of influences such as the indoor-outdoor temperature difference, wind speed or direction, and, in particular, occupant behaviors such as opening a window or running a kitchen fan [39,40,41,42]. 

The first source of error can be rectified by measuring indoor and outdoor concentrations. This is now being done for an ever-growing number of homes using low-cost particle sensors. The second source of error can only be dealt with statistically at present, due to the lack of low-cost personal monitors. The third source of error can be reduced by using the RCS model to identify periods when different infiltration factors may occur for a given location. This is the primary focus of our investigation.

We also present an improved method for determining PM_2.5_ from the raw data provided by the PurpleAir monitors. Although the Plantower manufacturer of the sensors used in the Purpleair monitors provides its own algorithms for determining PM_2.5_, no information is provided on the composition, density, or size distribution of the aerosol used for calibrating the sensors. Therefore we use an alternative algorithm based on the number of particles reported by the sensors. The algorithm is described fully in Section 2.3.

Finally, we employ a method of determining the limit of detection (LOD) of the PurpleAir monitors under real-life conditions. The LOD is of interest, particularly for indoor air, since it is a boundary below which the measured values are not significantly different from zero. Indoor air PM_2.5_ typically is at a lower concentration than outdoor air, and the fraction of particles below the LOD is typically higher than for outdoor air. The method is described fully in Section 2.4.

## 2. Materials and Methods

### 2.1. PurpleAir Monitors

PurpleAir monitors employ optical particle sensors manufactured by Plantower, a Chinese company. The monitors use one or two lasers at about 650 nm to scatter off the aerosols, which are brought into the monitor housing using a small fan. There is also a device providing the temperature and relative humidity (RH). PurpleAir recommends its PA-II monitor, which has two independent Plantower PMS-5003 sensors, for outdoor use, and its PA-I monitor, which has one PMS-1003 sensor, for indoor use. A photograph of a custom mounting of a PurpleAir CA-II outdoor monitor is provided (Appendix A). Using one of two proprietary Plantower algorithms, estimated values of PM_1_, PM_2.5_, and PM_10_ are provided. In addition, estimated particle numbers in six size categories are provided. All monitors send data to the Web, from which the observations can be downloaded either from the PurpleAir map page (https://www2.purpleair.com/, accessed on 1 January 2023) or the PurpleAir API site (https://api.purpleair.com/, accessed on 1 January 2023). All PurpleAir data used in this study employed an alternative algorithm (ALT-CF3) based on the particle numbers in the three smallest size categories. This algorithm is described fully in Section 2.3.

### 2.2. RCS Model

The RCS model assumes a constant infiltration factor over the time the house is monitored. The concentration due to particles of ambient origin is given by Equation (3) above. Suppose there is a set of average indoor concentrations and corresponding average outdoor concentrations. A regression of the indoor on the outdoor concentrations results in a regression line with slope *F_inf_*. Now a second line is drawn showing the contribution of outdoor particles to indoor PM_2.5_. That line is the solution to Equation (3) above for the outdoor-infiltrated particles only. The line is parallel to the slope of the regression and passes through the origin. Considering that the indoor contribution <*G*> cannot be negative, the line through the origin defines a “Forbidden Zone” where few observations are expected (Figure 1). For any given observation, the total indoor air concentration consists of two components: the outdoor-penetrated and indoor-generated particles. The outdoor-penetrated particles (vertical red line in Figure 1) are determined by the upper boundary of the Forbidden Zone. The indoor-generated particles (vertical blue line in Figure 1) are calculated by subtracting this value from the observed total indoor concentration.

A major weakness of the regression approach (including the RCS method) is the assumption of a constant infiltration factor, despite the influences mentioned above that alter the infiltration factor. For example, a major EPA study measured 37 homes in the Research Triangle Park of North Carolina for personal, indoor, and outdoor concentrations of sulfur and fine particles [11,17,18,19]. Each home was monitored for 7 days in each of the four seasons. The infiltration factor was considerably lower (0.49) in summer compared to the fall, winter, and spring values of 0.64, 0.62, and 0.59, respectively. The authors attributed this effect to the increased use of air conditioning during the summer. 

The existence of the Forbidden Zone is a feature that allows a check on the assumption of a constant infiltration factor. If many observations appear in the Forbidden Zone, the infiltration factor was not constant. However, it may then be possible to divide the dataset into two or more datasets, each with its own constant infiltration factor. 

This study deals with the problem of a variable infiltration factor. First, the RCS approach is applied to an entire dataset of matched indoor and outdoor daily average PM_2.5_ measurements. If there are multiple violations of the Forbidden Zone, subsets of the data are then sought that will have few or no violations. A rough estimate of the variability of the infiltration factor is the daily indoor/outdoor (I/O) ratio as a function of time. If there is a marked variation, it may suggest subsets of the data likely to have more stable ratios. For example, a seasonal variation (due to open windows in spring or the use of air conditioning in summer) might appear. If the resulting RCS regressions on each subset (or at least one of the subsets) show no or few violations of the Forbidden Zone, the result can be accepted as a robust estimate of the infiltration factor for that subset and thus a good estimate of the relative contributions of indoor-generated and outdoor-penetrated particles to the total indoor PM_2.5_.

The RCS approach is illustrated using matched indoor and outdoor long-term measurements (hundreds of days) at six California locations. The indoor sites are in Menlo Park, Oakmont (in Santa Rosa), Redwood City, and the Outer Sunset area in NW San Francisco. The outdoor sites are in those four locations and also at Bennett Valley, near Santa Rosa, and Alexander Avenue, in Redwood City.

#### 2.2.1. Menlo Park

This site has one outdoor PA-II monitor (with two Plantower PMS 5003 sensors) and one indoor PA-I monitor (with one Plantower PMS 1003 sensor). The monitors operated from 13 November 2019 to 4 December 2021. The site is a private residence occupied by a renter. Between 2 July 2020 and 14 February 2021, the renter was out of the country and the residence was unoccupied. Daily means were calculated for all days using the ALT-CF3 algorithm. At least 540 of the 720 possible measurements per day were required (18 h/day). For every two-minute measurement made by the outdoor monitors with two identical sensors A and B, a measure of precision (abs (A − B)/(A + B)) was calculated. Precision was required to be less than 0.2 to be included in the analysis. This resulted in the loss of about 8% of the data. The final database consisted of 270 days with matched indoor and outdoor daily means.

#### 2.2.2. Oakmont (in Santa Rosa)

Oakmont is a community within Santa Rosa, CA. The site is a home with two occupants. Two indoor PA-II monitors 1 and 2 were set up on 23 July 2019 and operated until 18 June 2020; at that point, two additional PA-II monitors (3 and 4) were deployed. Monitors 1 and 4 were operated exclusively indoors, and monitor 3 (except for one month) outdoors for the next 19 months to 28 September 2022. Daily means were calculated for all days using the ALT-CF3 algorithm. At least 360 of the 720 possible measurements per day were required. The final database consisted of 804 days with matched indoor and outdoor daily means.

#### 2.2.3. Redwood City

A site in Redwood City, CA operated two PurpleAir PA-II monitors indoors and one PurpleAir PA-II monitor outdoors from 1 May 2021 until 16 October 2022. The site is a home with one occupant. The final database consisted of 411 days with matched indoor and outdoor daily means.

#### 2.2.4. Alexander Avenue

Indoor and outdoor PurpleAir PM_2.5_ measurements at a residence on Alexander Avenue in Redwood City were collected between 5 December 2017 and 7 September 2022 using the ALT-CF3 algorithm. Daily means were calculated for all days with the requirement that at least half of the 720 possible measurements per day were available at an acceptable precision (abs (A − B)/(A + B)) of better than 20%. The final database consisted of 1645 days with matched indoor and outdoor daily means.

#### 2.2.5. Bennett Valley

Outdoor PurpleAir PM_2.5_ measurements at the Bennett Valley PurpleAir site in Sonoma County, CA were collected using the ALT-CF3 algorithm and compared to the Oakmont site in Santa Rosa (about 4 km distant) covering the period from 19 June 2020 to 27 September 2022. Daily means were calculated for all days with the requirement that at least half of the 720 possible measurements per day were available at an acceptable precision (abs (A − B)/(A + B)) of better than 20%. In total, 821 days met these requirements.

#### 2.2.6. Outer Sunset (San Francisco)

An outdoor and corresponding indoor site is located in the Outer Sunset area in northwest San Francisco. Data from 20 October 2019 to 13 August 2021 were examined. Daily means were calculated for all days with the requirement that at least half of the 720 possible measurements per day were available at an acceptable precision (abs (A − B)/(A + B)) of better than 20%. In total, 409 days met these requirements.

### 2.3. Algorithm for Calculating PM_2.5_

Although two algorithms (CF1 and CF_ATM) for calculating PM_2.5_ are offered by the Plantower (https://www.plantower.com/en/, accessed on 1 January 2023) manufacturer of the sensors used in the PurpleAir monitors, we reject both. There are several reasons for rejection: Both algorithms are biased high, from 40 to 90% [43,44,45].Both algorithms assign values of zero to some results below an arbitrary value. Yet there are no cases in which the reported number of particles is zero. This problem is particularly prevalent for indoor PM_2.5_ measurements, typically below outdoor levels. In one study with >900,000 indoor measurements, more than 200,000 values were reported as zero by the Plantower algorithms (Appendix A).No information is provided by Plantower regarding the calibration aerosol properties, such as the chemical makeup and its density.The precision of measurements is poor, in part because one or both sensors within the PA-II monitor often return zero values.The limit of detection (LOD) is elevated, again in part because of poor precision caused by excessive zero values.

The ALT-CF3 algorithm was developed to provide a transparent and reproducible alternative to the “black box” algorithms provided by Plantower. It is described fully in [46]. The algorithm uses the particle numbers reported for the three size fractions 0.3–0.5 µm, 0.5–1 µm, and 1–2.5 µm. By assigning an average diameter to each size fraction, the total particle volume is calculated. In our approach, we selected as an average diameter the geometric mean of the upper and lower boundaries of these three size categories. The total PM_2.5_ mass is then determined in two steps: first, a density is assigned (in this case the density of water) and then a calibration factor is determined by comparison to nearby regulatory monitors employing Federal Reference or Federal Equivalent Methods (FRM/FEM). The original calibration factor for 33 PurpleAir PA-II monitors within 0.5 km of 27 regulatory monitors was 3.0, leading to the naming of this alternative algorithm ALT-CF3. The equation giving the ALT-CF3 estimate of PM_2.5_ is
PM_2.5_ = 3(0.00030418 N1 + 0.0018512 N2 + 0.2069706 N3)(5)
where N1, N2, and N3 are the number of particles per deciliter in the three smallest size categories. 

The ALT-CF3 method has been found to provide less bias, better precision, and a lower limit of detection than either the CF1 or CF_ATM algorithms [47,48]. This method is presently available on the PurpleAir mapping page at the PurpleAir Website (https://www2.purpleair.com/, accessed on 1 January 2023) as an alternative to the proprietary algorithms offered by Plantower. The method is also available on the PurpleAir API page (https://api.purpleair.com/, accessed on 1 January 2023) under a different name: “PM_2.5__alt”. All PM_2.5_ data in this paper were determined using the ALT-CF3 algorithm. 

### 2.4. Calculation of the Limit of Detection (LOD)

A method for calculating the limit of detection for continuous monitors has been developed [49]. By analogy with integrated methods such as weighing particle masses on filters, it is required that the mean of multiple collocated instruments of one type all measuring the same environment at some low concentration exceeds 3 times the standard deviation to be considered as evidence at the 95% confidence level of a non-zero concentration. This approach can be applied to a single PurpleAir monitor with two sensors A and B. Every data point has an associated standard deviation based on the two sensors. The data can be ordered by the mean of the two sensors. At very low mean values, it is likely that the ratio of the mean to the standard deviation (µ/σ) will be <3 for many of the observations. For higher mean concentrations, the fraction of cases with µ/σ < 3 will decline. When this fraction falls below 5%, the associated mean concentration will be the LOD. Operationally, this approach can be implemented by considering successive batches of, say, 100 measurements of the ordered data. When the number of ratios <3 falls to fewer than 5 of the 100 values in the batch, the associated mean of that batch may be considered a provisional LOD. However, a batch at a higher concentration may have 5 or more ratios <3, in which case the LOD will be greater than the provisional LOD and the search for the LOD must continue to higher concentrations. This problem can be solved by analyzing all possible batches of 100 and picking the highest concentration with only 5 out of 100 values having µ/σ < 3.

In our study, however, it was found that some of the very large multiple-day datasets, including 100,000 to 975,000 measurements, gave clearly erroneous high values for the LOD, likely due to random variation. We therefore tweaked the method to employ “batches” of 1000 (instead of 100), with 50 (instead of 5) µ/σ values <3 to constitute the 5% cutoff for determining the LOD. This approach was sufficient to smooth out the function and allow the identification of the LOD. The approach is described in an example in the Alexander Avenue section of the Appendix A. 

## 3. Results

### 3.1. Menlo Park

The data for the time of occupancy (270 days) consist of 372,621 two-minute average PM_2.5_ concentrations (Table 1). Note the loss of nearly 70,000 data points for the Plantower CF1 algorithm (bottom three rows) compared to the ALT-CF3 algorithm (top five rows). This is partly due to the Plantower decision to assign values of zero to concentrations below an arbitrary cutoff. No values of zero ever occur in the ALT-CF3 algorithm, since there are never occasions when the number of particles between 0.3 and 0.5 µm is zero. The CF1 algorithm overestimates the outdoor PM_2.5_ concentration by about 85% (8.3 vs. 4.4 µg/m^3^) and the indoor concentration by 50% (3.3 vs. 2.2 µg/m^3^). The mean PM_2.5_ concentration due to indoor activities (1.78 µg/m^3^) is larger than the mean concentration (0.78 µg/m^3^) attributable to penetration of ambient PM_2.5_. The Spearman correlation coefficient was reasonably high at 0.67.

For the 270-day period when the residence was occupied, the RCS estimate of the infiltration factor is 0.0849 (Figure 2). The RCS-predicted Forbidden Zone created by the product of *F_inf_* and the outdoor concentration is not violated by a single daily measurement. This suggests that the estimate of a single constant value for the infiltration factor throughout the entire period is robust. 

#### Limit of Detection (LOD) at Menlo Park

Using the approach to calculate the LOD described in [49], the ALT-CF3 algorithm resulted in an estimated LOD of 1.10 µg/m^3^, with 73,458 (16.6%) PM_2.5_ ALT-CF3 values below the LOD. For the Plantower CF1 algorithm, the LOD was almost four times higher at 4.15 µg/m^3^ and nearly half (48.6%) of all values fell below the LOD.

### 3.2. Oakmont, Santa Rosa, CA

A table of results for the Plantower algorithms CF1 and CF_ATM for comparison with the ALT-CF3 algorithm is provided (Table 2). The Plantower algorithms overestimated PM_2.5_ values by 50% (CF1) and 30% (CF_ATM) compared to the ALT-CF3 algorithm, which has been calibrated against multiple regulatory monitors [30]. The Plantower algorithms resulted in a major loss of data (18%) due to the Plantower practice of assigning values of zero to results below an arbitrary cutoff. (The ALT-CF3 algorithm resulted in no zero values.) The mean precision of 22% for both Plantower algorithms is nearly three times the mean of 8% shown by the ALT-CF3 algorithm.

Quality assurance of the data was performed by setting a limit of 0.2 (20%) on the precision of both the indoor and outdoor measurements. This resulted in ~9% of the ALT-CF3 data and 22% of the CF1/CF_ATM data being deleted. In addition, several experiments resulting in elevated PM_2.5_ for periods of several hours in one room of the house (which included at least one of the indoor monitors) were removed from consideration. This resulted in a further removal of ~0.6% of the data. The final database consisted of 804 days with matched indoor and outdoor daily means.

At the Oakmont site, monitors 1 and 4 were always indoors, monitor 2 was mostly indoors, and monitor 3 was outdoors except for one month. The total number of zeros reported by the Plantower CF1 (and CF_ATM) algorithm for PM_2.5_ was >100,000 (17–22%) for the two indoor monitors and less than about 40,000 (6–7%) for monitor 3, which was mostly outdoors (Table 3). This indicates how the lower concentrations found indoors lead to increased numbers of zeros reported by the Plantower algorithms compared to outdoors.

#### Limit of Detection (LOD) for Oakmont

The LOD was calculated for each monitor location for both the Alt-CF3 and Plantower CF1 algorithms (Table 4). For the ALT-CF3 algorithm, the LOD ranged from 0.60 to 1.32 µg/m^3^ and was always exceeded by a majority of the observations. For the Plantower CF1 algorithm, the LOD ranged from 2.9 to 9.9 µg/m^3^, and fewer than half the observations exceeded the LOD except for a one-month period when monitor 3 was moved indoors.

The regression of the indoor on outdoor PM_2.5_ using the ALT-CF3 algorithm is provided (Figure 3). However, there are 120 of 804 (15%) values in the Forbidden Zone, suggesting that the estimate of the infiltration factor is not robust. 

Analysis of the monthly regressions showed a clear pattern of low infiltration factors (mean 0.11, SE 0.02) for the cool wet months of December through May (227 days), and high infiltration factors (mean 0.37, SE 0.04) for the warm dry months of June through November (285 days) (Figure 4). The 6 months (June-November) with high infiltration factors and 6 months (December through May) with low infiltration factors match up very well both with temperature and rainfall. The 6 months with the highest average temperatures in Santa Rosa (59–66 °F) are May–October, while the 6 months with the lowest temperatures (46–55 °F) occur in November–April. Rainfall is also limited to the November–April time frame, with the six months from May to October seldom experiencing rain. Doors and windows are more likely to be open when the weather is warm and dry, resulting in a higher infiltration factor in the warm months. The I/O values appear to be a sine wave with a period of one year, showing peaks in the summer months and troughs in the winter months.

As a result, the data were split into two 6-month periods. The 6-month period from December to May had few points falling into the Forbidden Zone (Figure 5). Therefore the estimated infiltration factor of 0.1355 is accepted, resulting in an average contribution of 1.54 µg/m^3^ made by indoor-generated particles during this period.

During the remaining 6 months of June–November, once again few points fall into the Forbidden Zone (Figure 6). The estimated infiltration factor was 0.340, more than twice as high as the winter value of 0.1355, and the estimated contribution of indoor-generated particles was 2.09 µg/m^3^.

### 3.3. Redwood City, CA

A site in Redwood City, CA operated two PurpleAir PA-II monitors indoors and one PurpleAir PA-II monitor outdoors. The 2 min average indoor data for the period from 1 May 2021 until 16 October 2022 are provided for both the Plantower CF1 and ALT-CF3 algorithms in Table 5. The Plantower CF1 algorithm overestimated PM_2.5_ by about 50% (6.2 vs. 4.1 µg/m^3^). The mean precision of 17–21% in the CF1 algorithm was about twice the 9–10% observed for the ALT-CF3 algorithm. Applying an upper bound of 20% to the precision resulted in about 15% of the PM_2.5_ CF3 data being removed from the analysis, compared to 40% of the Plantower CF1 data. 

The extreme loss of data from the use of the Plantower CF1 algorithm is largely due to the practice of setting all PM_2.5_ values falling below a cutoff to zero. The number of zeros reported by the Plantower CF1 algorithm ranged from 49,013 to 72,442 (7–11% of all data). No zeros are ever reported by the ALT-CF3 algorithm, since particles are always present in the 0.3–0.5 µm size category.

Daily means were calculated for all days with the requirement that at least half of the 720 possible measurements per day were available at an acceptable precision (abs (A − B)/(A + B)) of better than 20%. The final database consisted of 411 days with matched indoor and outdoor daily means.

The RCS approach for the entire database results in 120 (29%) values in the Forbidden Zone, indicating that the infiltration factor varies too much to be estimated using the entire database (Figure 7).

The I/O ratio is shown for the entire period (Figure 8). It suggests a sine wave with a period of one year, with peaks occurring in the summer months and troughs in the winter months. A period of low infiltration factors appears between about November and March. 

The RCS regression on the November–March period results in only 15 points falling into the Forbidden Zone, and 12 of these seem to “hug” the line (Figure 9). Therefore the estimated infiltration factor of 0.2329 and mean contribution of 0.556 µg/m^3^ made by indoor-generated particles during this period appears robust.

An effort was made to estimate the infiltration factor for the summer months (June–July–August). However, the effort failed, suggesting too much variability of the infiltration factor. Two of these months were strongly affected by wildfires. 

#### Limit of Detection (LOD)

For monitor 1 indoors, the LOD for the ALT-CF3 algorithm was about 1.4 µg/m^3^, corresponding to about 97% of all observations exceeding the LOD. For the same monitor, the Plantower CF1 algorithm resulted in an estimated LOD of 4.75 µg/m^3^, with only 42% of observations exceeding the LOD.

### 3.4. Alexander Avenue

The I/O ratio is shown for the entire period between 5 December 2017 and 7 September 2022 (Appendix A). The final database consisted of 1625 days with matched indoor and outdoor daily means. Unlike our previous findings with homes in Santa Rosa and Redwood City, the I/O ratio shows little or no seasonal variation. There is, however, an apparent sudden switch to less variable values occurring near January 2020. 

The RCS approach to estimate the infiltration factor for the entire database resulted in 516 of 1625 (32%) values in the Forbidden Zone (Figure 10). 

Based on the apparently smaller variability of the I/O ratios in the latter part of the period from about 1 January 2020 to 7 September 2022 (938 days), the RCS approach is applied to that period (Figure 11). In this period, 75 (<8%) of the observations fall into the Forbidden Zone. This is somewhat marginal. However, if applicable, it would suggest an infiltration factor of 0.08 and an average contribution of about 0.51 µg/m^3^ from indoor-generated particles. The contribution of particles of ambient origin was 0.44 µg/m^3^, so the indoor-generated fraction of the total indoor potential exposure was about 54%.

An effort was made to estimate the infiltration factor for the period from 2017 through 2019. However, this effort was unsuccessful. 

For this location, there appears to have been a period before 2020 with a higher infiltration factor and a period from 2020 to the present with a lower infiltration factor. Only the latter period allowed the finding of a likely stable infiltration factor and therefore an estimate of the relative importance of indoor-generated and outdoor-penetrated particles. 

### 3.5. Bennett Valley

The matched Bennett Valley outdoor and Oakmont indoor measurements numbered 821 days However, an unusually high number (280) of the Bennett Valley measurements showed precision poorer than 20%. Only 21 of the 821 corresponding Oakmont measurements had precision poorer than 20%. The final database consisted of 522 days with matched indoor and outdoor daily means and precision better than 20%.

The I/O ratio is provided for the entire period (Appendix A). There are two strong troughs in winter 2021 and 2022, with peaks in the summer of 2020 and 2021. 

The RCS approach to estimate the infiltration factor for the entire database results in 172 of 522 (33%) values in the Forbidden Zone, indicating that the infiltration factor varies too much to be estimated using the entire database (Figure 12).

The variation of the I/O ratio suggests that a period of low infiltration factors might occur in the winter and perhaps the spring. Indeed, when the RCS regression is performed on the winter months (December through February) and the spring months (April–June), few points fall into the Forbidden Zone (Figure 13). Therefore the estimated infiltration factor of 0.2195 appears robust. An average contribution of 1.1 µg/m^3^ is made by indoor-generated particles.

The corresponding regression on the summer–fall months (August–November) again produced few observations violating the Forbidden Zone, with an estimated infiltration factor of 0.3637 and an estimated contribution of indoor-generated particles of 2.10 µg/m^3^ (Figure 14).

Plotting the indoor-outdoor ratio (I/O) vs. time was useful in showing the seasonal change in the infiltration factor. Two periods (winter–spring and summer–fall) were found to have a stable infiltration factor, allowing good estimates of both the infiltration factor and the relative contribution of indoor-generated and outdoor-penetrated particles to total indoor potential exposure. The good results for sites 4 km apart are notable, suggesting that outdoor sites may often have little spatial variation. This opens up the possibility of using distant outdoor sites (e.g., those with high-quality Federal Reference Method (FRM) PM_2.5_ measurements) to calculate infiltration factors at many indoor sites. 

### 3.6. Outer Sunset (San Francisco)

Data from 20 October 2019 to 13 August 2021 (409 days) were examined (Figure 15). This site in San Francisco had very few points falling in the Forbidden Zone. The intercept of 10.3 µg/m^3^ is an unusually strong estimate of the dominance of the (unknown) indoor source(s).

## 4. Discussion

The LODs for five of the six sites are provided for both the ALT-CF3 and CF1 algorithms in Table 6. The mean CF1 LOD of 4.24 µg/m^3^ was four times larger than the ALT-CF3 LOD of 1.04 µg/m^3^. Moreover, the lowest of 11 values found for the CF1 algorithm was 2.69 µg/m^3^, while the highest value for the ALT-CF3 algorithm was 1.83 µg/m^3^, indicating no overlap between the two algorithms.

Many investigators have estimated LODs for a number of low-cost monitors. Many of these are chamber studies, in which they can compare the low-cost monitors to a reference instrument at a very low concentration (e.g., <1 µg/m^3^). Chamber studies have shown low LODs of <1 µg/m^3^ to about 2 µg/m^3^ [50,51]. A laboratory investigation used the method in [49] to calculate the LOD for five low-cost monitors [51]. All monitors were below 1 µg/m^3^. The PurpleAir monitor had the lowest LOD (near zero!).

Typically, chamber studies under careful control of the particle composition, and perhaps size, temperature, humidity, and a nearby reference instrument, show lower LODs than field investigations. For example, a field and laboratory study of the PMS 1003 sensor found that it ranged from 1 to 3.22 µg/m^3^ under laboratory conditions but increased to 10.5 µg/m^3^ under ambient conditions [52].

Our method for determining the LOD under field conditions has been adopted by several investigators [51,53]. One study included 58 Speck monitors during two outdoor and one indoor 3-day campaigns and found a LOD for 1 min data of 10 µg/m^3^, or 9 µg/m^3^ for hourly average data [53]. In 2016–2017, a field study of two Plantower MS 1003 sensors took place over 320 days, with the addition of a 5003 sensor in 2017 [54]. In 2017, the PMS LODs ranged from 2.62–11.5 µg/m^3^. This range is very similar to our range of 2.69–9.9 µg/m^3^ using the Plantower algorithms.

It may not be well understood how low typical exposures to PM_2.5_ are. Comparing the mean LOD of 4.2 for the CF1 (or CF_ATM) algorithm to actual measured levels of PM_2.5_ in a tristate area over the last 4.7 years, it is well above the median value for indoor hourly average PM_2.5_ concentrations of 3.5 µg/m^3^ [55]. This means that the use of the Plantower algorithm would result in the conclusion that well over half of the indoor daily average PM_2.5_ concentrations experienced by persons in ~9000 residences over 4.7 years would not be significantly different from zero. Use of the ALT-CF3 LOD of 1 µg/m^3^ results in only about 15% of the measured population exposures falling below that figure. As mentioned in the Materials and Methods section, we found that our original method for determining the LOD, which employed inspecting the µ/σ ratios in batches of 100 contiguous concentrations, failed for some of the datasets. By changing to batches of 1000, we reduced the noise sufficiently to achieve a reasonably smooth function (Figure 16).

A summary of the estimated preliminary and final infiltration estimates, along with the calculated outdoor-infiltrated and indoor-generated concentrations for all six cases is offered (Table 7).

In Table 7, the values shown in red represent the four cases in which the Forbidden Zone was violated by numerous observations. These results have been discarded and replaced by the values shown in the bottom eight rows for subsets of the data with many fewer violations of the Forbidden Zone.

In two cases, (Menlo Park and Outer Sunset), the number of observations violating the Forbidden Zone were near zero, and thus provided reassurance that the entire data set (last two columns, top four rows) could be used to determine robust estimates for the infiltration factor and also for calculating the contributions to total indoor PM_2.5_ concentration of indoor-generated and outdoor-penetrated particles. In these two cases, the indoor-generated particles greatly outweighed (80–85%) particles of ambient origin. In the remaining four cases, the regression on the full dataset failed to identify a stable infiltration factor (red typeface, first four columns, top four rows). For two of these four cases (Oakmont and Bennett Valley), the use of the I/O ratio identified two 6-month periods (subsets 1 and 2, first two columns, bottom eight rows) that could be used to calculate robust estimates of the infiltration factor for each period. In the remaining two cases (Redwood City and Alexander Avenue), it was possible to identify one subset with a stable infiltration factor (columns 3–4, rows 5–8), although the remaining subset could not be quantitatively analyzed.

The importance of determining the correct infiltration factors can be seen by considering the first column in Table 7 for Oakmont. Using the entire dataset (with the many violations of the Forbidden Zone), the contribution of indoor-generated particles was 34%, indicating that the outdoor-infiltrated particles (66%) were about twice as important. However, using the two subsets 1 and 2 with good estimates of the infiltration factor shows that the actual contribution of the indoor-generated particles was 48% in one case and 80% in the other, an average share of 64% or about twice as important as the particles of ambient origin (36%).

This example leads to the following general conclusion: When a regression of indoor on outdoor fine particles leads to multiple violations of the Forbidden Zone, the contribution of indoor-generated particles to the total indoor concentration may be underestimated. This is illustrated using the example of the Oakmont site. By coloring the winter–spring (Dec–May) and summer–fall (June–November) measurements differently (combining Figure 3, Figure 5 and Figure 6), the violations of the Forbidden Zone for all data combined (points below the upper boundary given by the equation y = 0.3437x) can be seen to consist mostly of the winter–spring measurements (blue) (Figure 17). When these data are isolated, the resulting slope of 0.1355 is almost a factor of three below the slope of the regression on all datapoints, meaning the influence of particles of ambient origin is reduced by the same factor. Or, in other words, the influence of indoor-generated particles on total indoor exposure is increased by the same factor.

One recent study has calculated long-term exposures for more than 3000 PurpleAir indoor sites in three states over a 4.7-year period using RCS regressions on about 10,000 outdoor sites and found that indoor-generated particles make about the same contribution to total indoor exposures as particles of ambient origin [55]. Presuming that some of these regressions resulted in violations of the Forbidden Zone, the conclusion that indoor-generated particles constitute about half of all potential indoor exposure may be an underestimate of the actual contribution.

The Alt-CF3 algorithm outperformed the Plantower CF1 algorithm in every aspect considered. Precision was better, the limit of detection was lower, the number of values > LOD was higher, and no zero values were recorded compared to hundreds of thousands of zeros for the Plantower CF1 algorithm. Although the Plantower CF_ATM algorithm was not directly considered in all analyses, it is identical to the Plantower CF_1 algorithm for all values less than ~28 µg/m^3^, and since these constitute the majority of observations, and include all the zeros reported by Plantower CF_1, the findings regarding precision, LOD, and number of zeros for the CF_1 algorithm would be equally applicable to CF_ATM. Since all the alternative algorithms (or “conversion factors”) other than ALT-CF3 offered by PurpleAir depend on one or another Plantower algorithm, they are all equally vulnerable to the Plantower shortcomings.

In Oakmont, Santa Rosa, a PurpleAir PA-II monitor measured indoor concentrations of PM_2.5_ continuously from 1/10/19 to 8/21/22, collecting 988,825 eighty-second or two-minute averages. The data were analyzed using five conversion factors made available by PurpleAir on their map page or their API link. The conversion factors include the CF1 algorithm provided by Plantower, the ALT-CF3 algorithm (also described as “PM_2.5_ alt” on the API site) [30,46], the EPA model [44], the LRAPA model used in Lane County, Oregon, (https://www.lrapa.org/, accessed on 1 January 2023), and the Australian woodsmoke model [56]. A sixth conversion factor is not presently available from PurpleAir [45]. Five of the six algorithms agree well at high PM_2.5_ concentrations (Figure 18).

Only the CF1 algorithm is out of step, reading about 90% higher than the other five models. At low concentrations, however, all algorithms except the ALT-CF3 algorithm are unable to chart the actual concentrations due to the CF1 assignment of a value of zero to these concentrations. The number of zeros reported by the CF1 algorithm was 205,682 for sensor A and 204,996 for sensor B, or about 21% of all data being lost by using the Plantower algorithms. Since all the models shown other than the ALT-CF3 model depend on the CF1 or CF_ATM algorithm, they are all affected by the Plantower arbitrary assignment of values to zero. Three models include an intercept that provides nonzero estimates for all cases, but these estimates do not seem to be based on real data. The ALT-CF3 model, however, suggests that these lower 20% of concentrations are in fact measurable and represent a reasonable continuation of the roughly log-normal distribution shown for the other 80% of the data.

## 5. Conclusions

Epidemiological studies without indoor measurements are unable to arrive at estimates of indoor-generated particles, yet some of these particles can have health effects. The advent of low-cost particle sensors can improve our knowledge of indoor potential exposure to PM_2.5_. These sensors make it possible for the first time to estimate long-term (months to years) PM_2.5_ concentrations from indoor sources. The present study has shown how to use the Forbidden Zone to evaluate the quality of the estimate of the infiltration factor. In particular, the use of the simple I/O ratio can often suggest subsets of the data to analyze with a chance of identifying periods with a stable infiltration factor, thus providing a more accurate determination of both indoor-generated and outdoor-penetrated PM_2.5_. Finally, a recently developed algorithm for calculating PM_2.5_ from PurpleAir measurements was found to be superior to the proprietary algorithms offered by the manufacturer of the sensors used in PurpleAir monitors. The use of this algorithm reduced the limit of detection by a factor of four compared to the LOD as calculated using the proprietary Plantower algorithms.

Practical Implications

A simple method is developed to determine whether PM_2.5_ infiltration factors found by regression are reliably constant. If not, the method can help identify subsets of the data that may have constant infiltration factors.The method is illustrated by using long-term (hundreds of days) indoor and outdoor PM_2.5_ measurements made by PurpleAir low-cost particle monitors.An algorithm for calculating PM_2.5_ is used that avoids entirely the proprietary algorithms of the manufacturer of the Plantower sensors used in PurpleAir monitors. This algorithm (ALT—CF3) is shown to have reduced bias, increased precision, and much lower limits of detection (LOD).An improved method of determining the LOD of any continuous particle monitor using only real-life (field) data is presented.

## Figures and Tables

**Figure 1 sensors-23-01160-f001:**
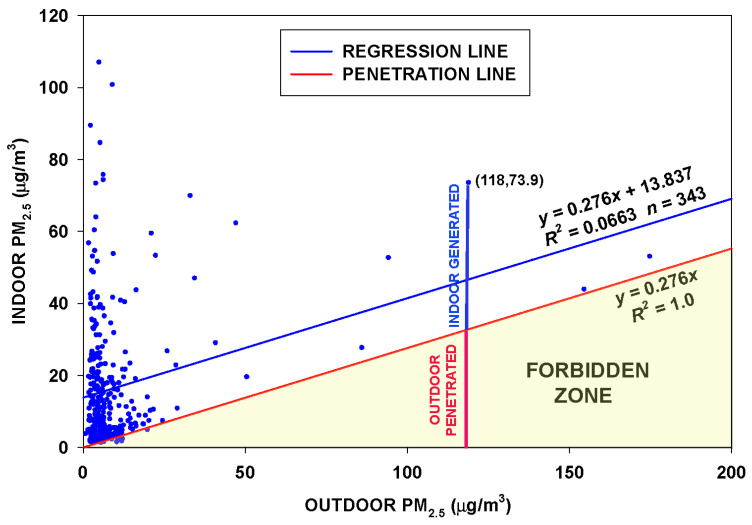
Regression of indoor on outdoor PM_2.5_ measured over 343 days for a home. The slope of the regression line (blue) is the infiltration factor *F_inf_*. The parallel line (red) is the product of the infiltration factor *F_inf_* and the outdoor concentration. This line forms the upper boundary of a “Forbidden Zone”. No observed point should appear in this zone if *F_inf_* is constant and applies to all data, because it would indicate a negative indoor-generated concentration.

**Figure 2 sensors-23-01160-f002:**
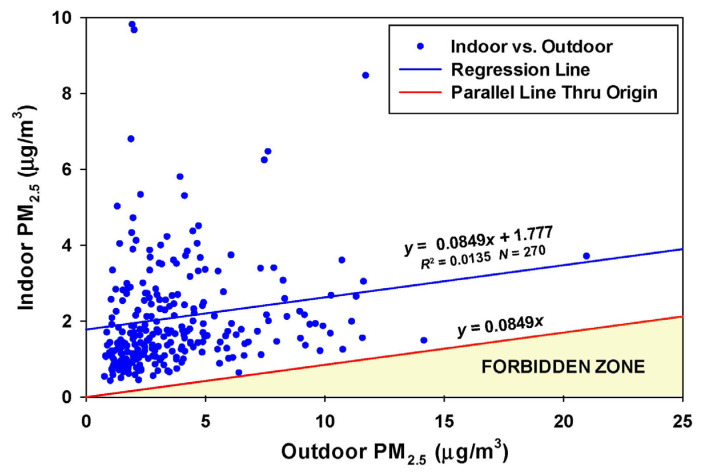
Regression of daily outdoor PM_2.5_ on indoor PM_2.5_ during the time the Menlo Park residence was occupied (13 November 2019 to 2 July 2020 and 14 February 21 to 4 December 2021).

**Figure 3 sensors-23-01160-f003:**
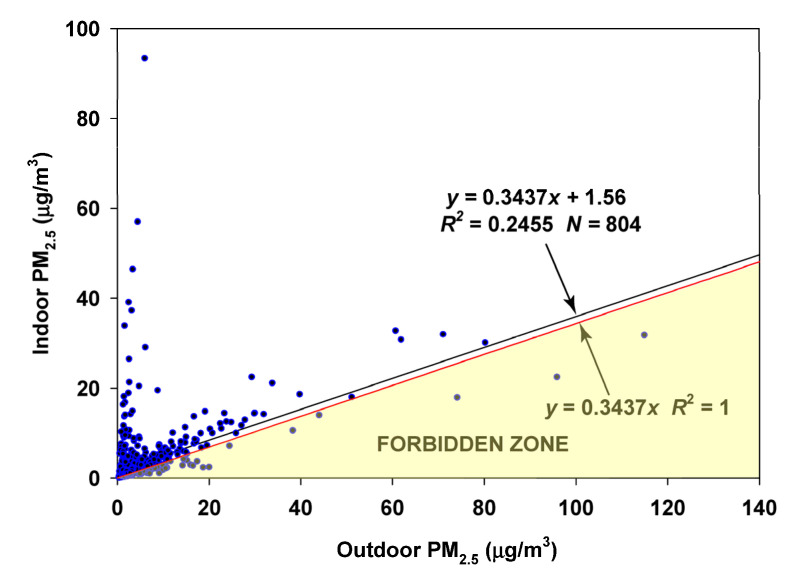
Random Component Superposition (RCS) regression for Oakmont.

**Figure 4 sensors-23-01160-f004:**
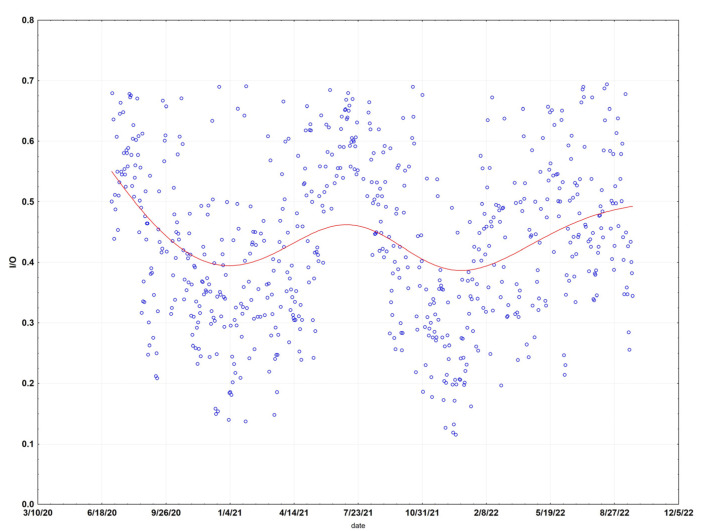
Indoor-outdoor (I/O) ratio of daily means at the Santa Rosa site. Data have been fit (red line) using distance-weighted regression.

**Figure 5 sensors-23-01160-f005:**
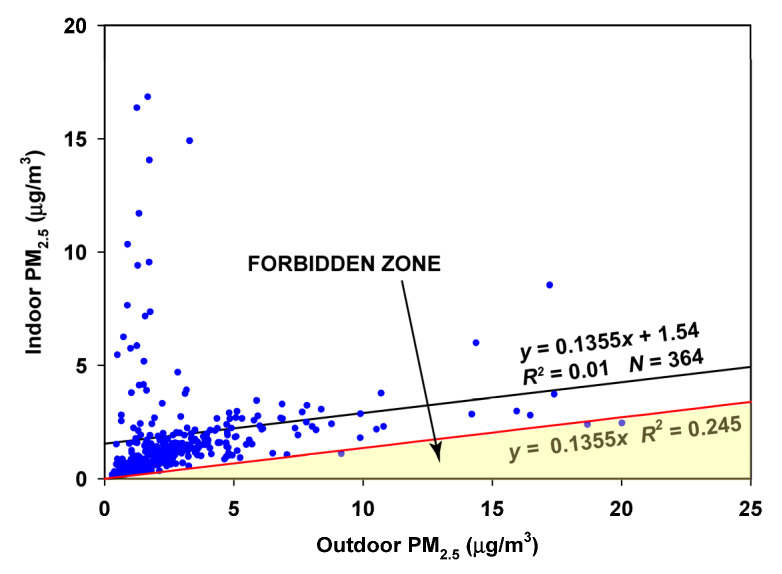
RCS regression of indoor on outdoor air for the December through May data in Oakmont.

**Figure 6 sensors-23-01160-f006:**
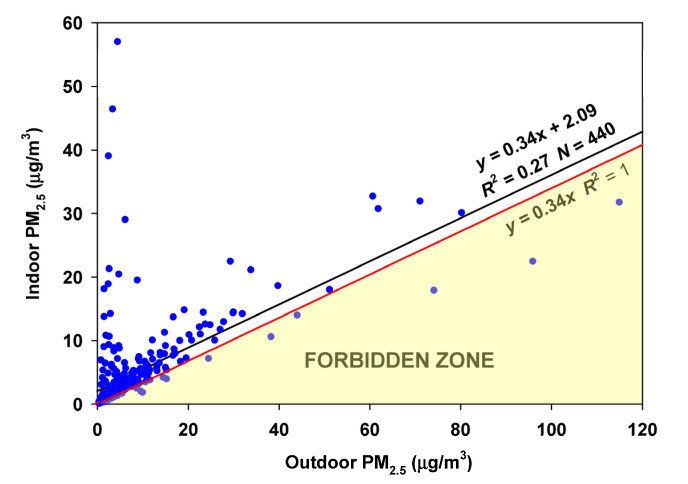
RCS regression of indoor on outdoor air for the June through November data in Oakmont.

**Figure 7 sensors-23-01160-f007:**
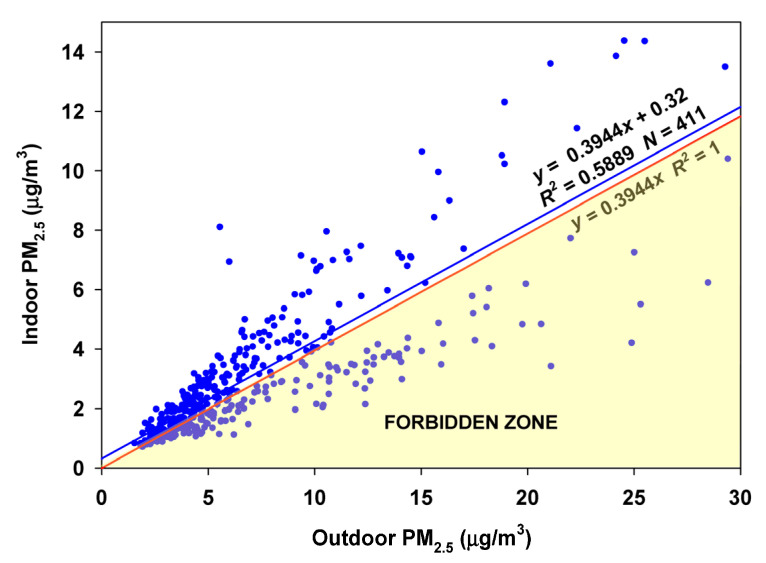
Random Component Superposition (RCS) regression for the Redwood City site. The slope (*F_inf_*) of 0.3944 results in many datapoints falling into the Forbidden Zone (below the estimated outdoor-penetrated particles given by the red line).

**Figure 8 sensors-23-01160-f008:**
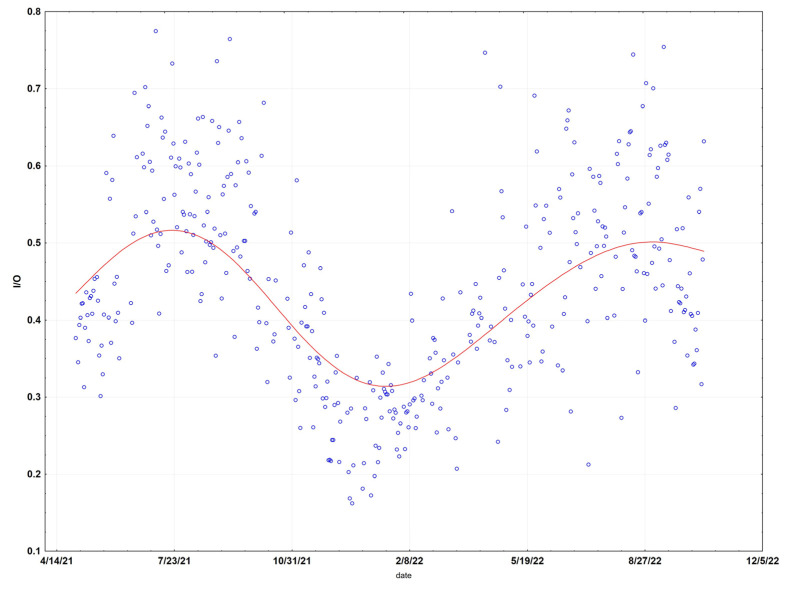
Indoor-outdoor (I/O) ratio of daily means at the Redwood City site.

**Figure 9 sensors-23-01160-f009:**
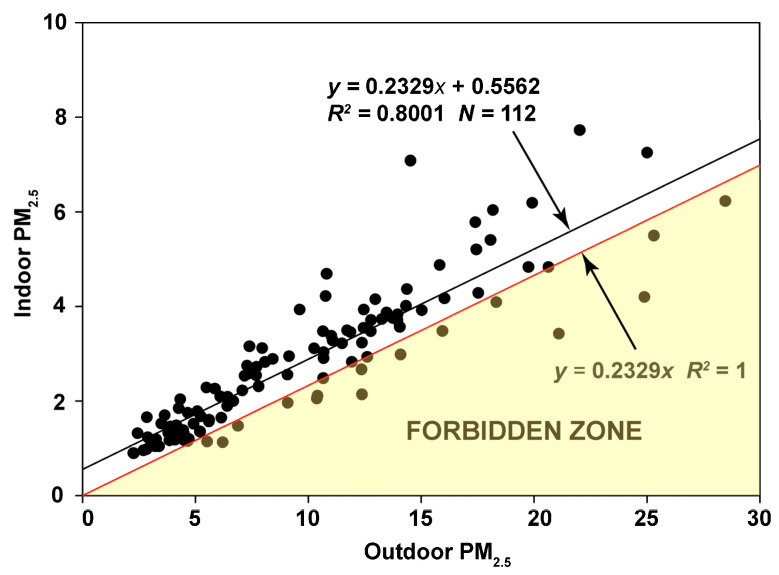
Random Component Superposition (RCS) regression for the November through March data at Redwood City.

**Figure 10 sensors-23-01160-f010:**
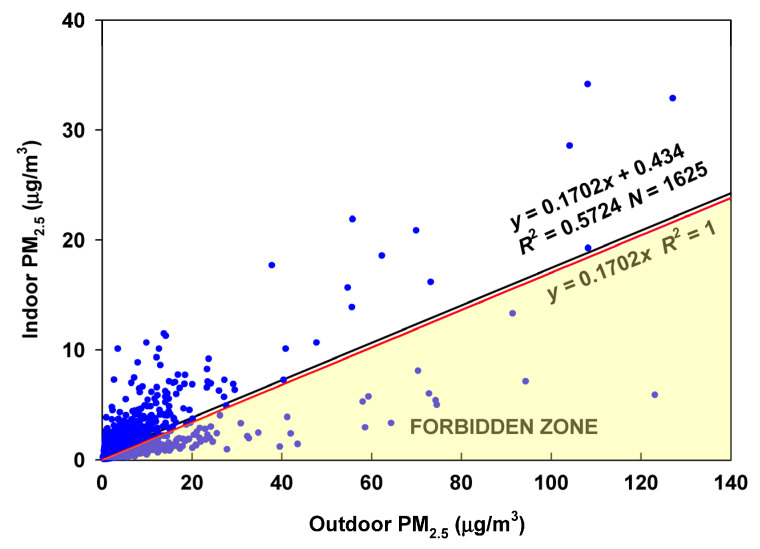
Regression of Alexander Avenue indoor PM_2.5_ on Alexander Avenue outdoor PM_2.5_.

**Figure 11 sensors-23-01160-f011:**
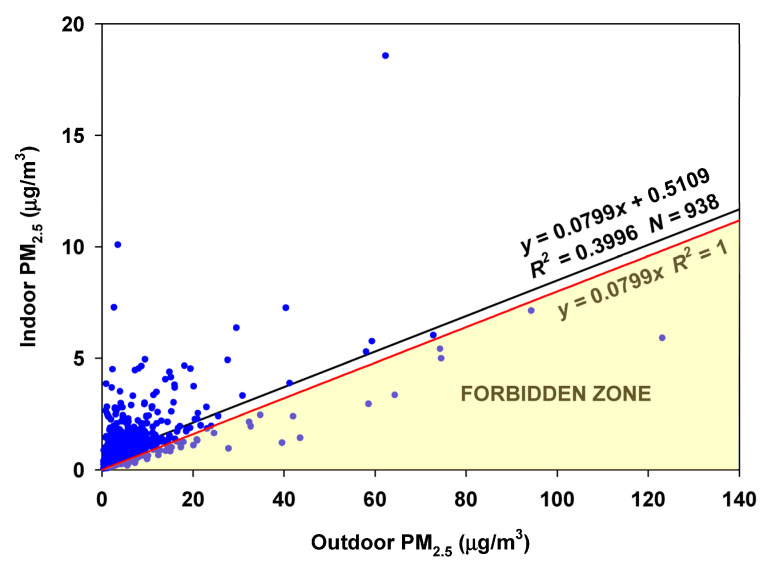
RCS regression for the period from 2020 to 2022.

**Figure 12 sensors-23-01160-f012:**
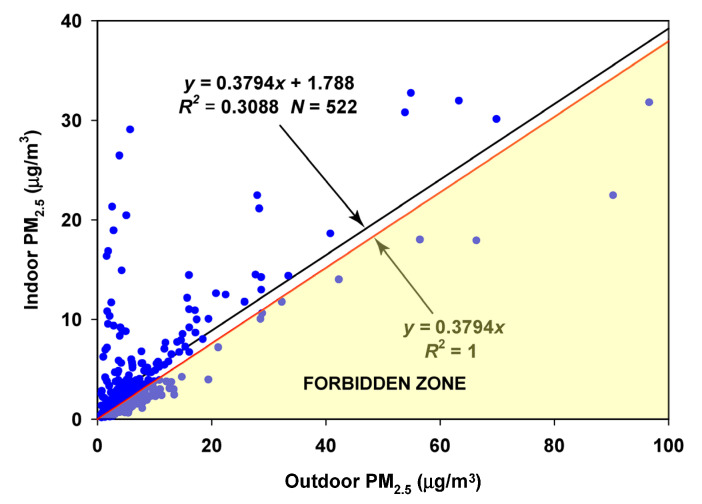
RCS regression of indoor values at Oakmont on outdoor values at Bennett Valley, about 4 km from Oakmont.

**Figure 13 sensors-23-01160-f013:**
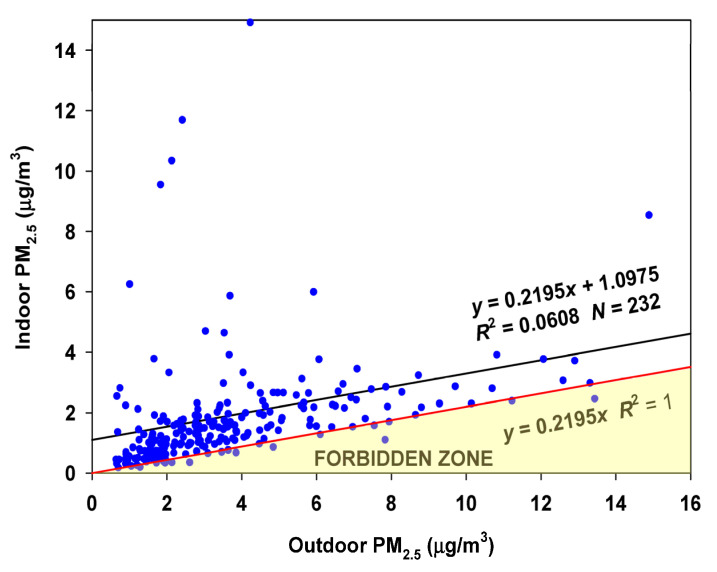
Regression of Oakmont indoor concentrations on Bennett Valley outdoor concentrations for the winter and spring months (December–June (March omitted)).

**Figure 14 sensors-23-01160-f014:**
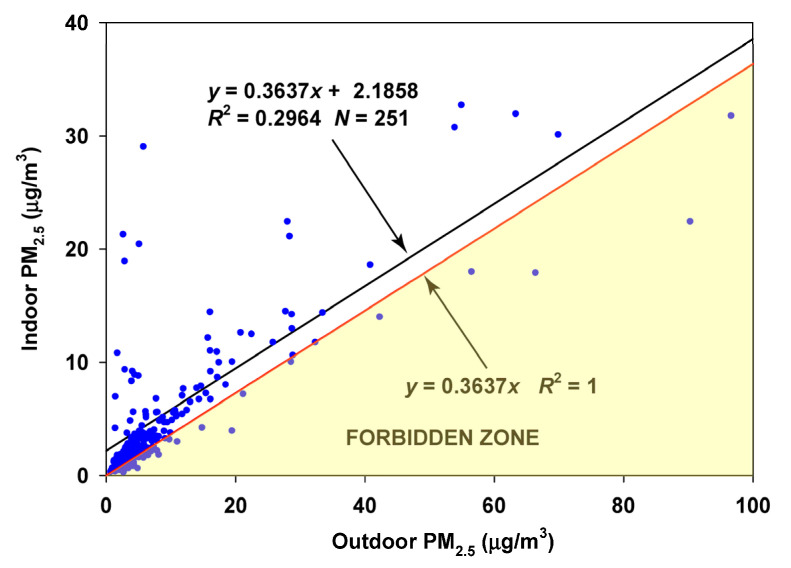
Regression of Oakmont indoor concentrations on Bennett Valley outdoor concentrations for the summer and fall months (August–November).

**Figure 15 sensors-23-01160-f015:**
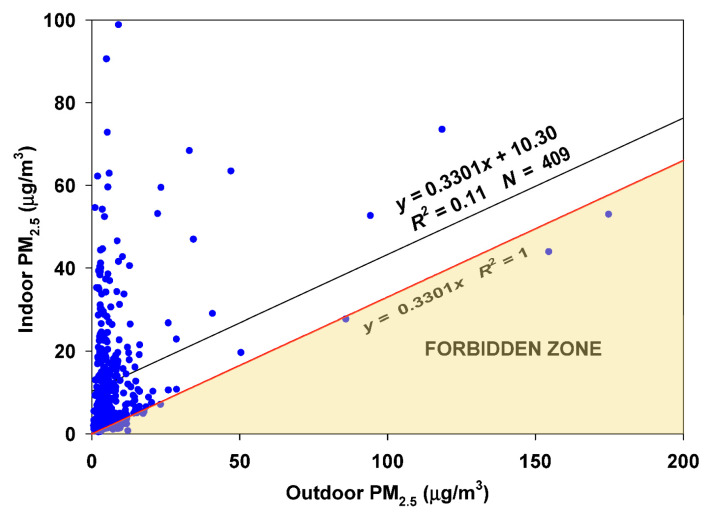
RCS regression of indoor on outdoor data at two nearby sites in the Outer Sunset area of San Francisco.

**Figure 16 sensors-23-01160-f016:**
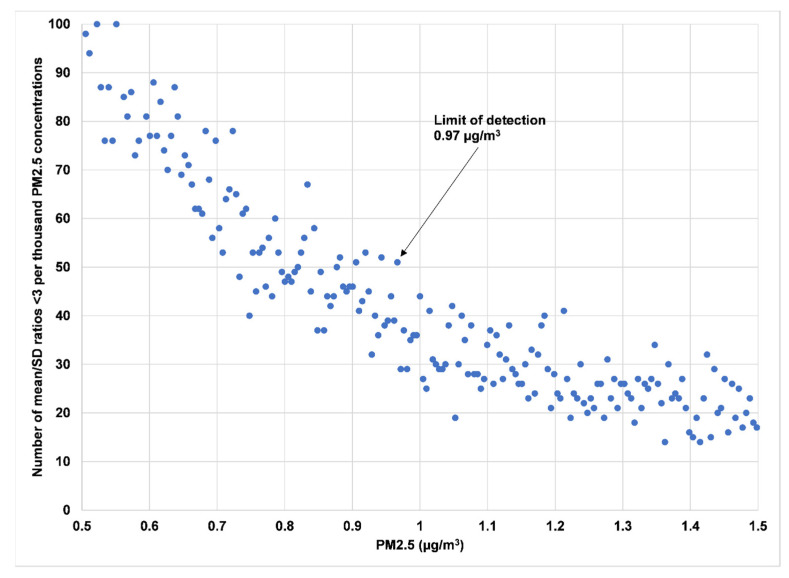
The number of mean/SD ratios per thousand “batches” of ordered contiguous measured concentrations steadily declines with increasing concentration. When this function falls below 50 and does not exceed 50 at any higher concentration, the LOD is considered to be the “last” (largest) concentration with at least 50 ratios <3.

**Figure 17 sensors-23-01160-f017:**
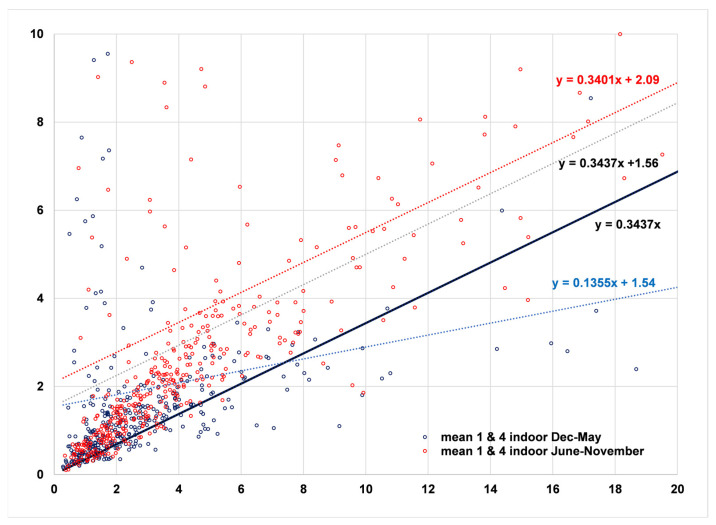
Regressions of Oakmont indoor PM_2.5_ measurements on outdoor values. The equation for the combined data is y = 0.3437x + 1.56 µg/m^3^ (light black dashed line) and the resulting upper boundary of the Forbidden Zone is y = 0.3437x (thick black line passing through the origin). The equation for the summer–fall data (red) is 0.3401x + 1.56. The equation for the winter–spring data (blue) is 0.1335 + 1.54.

**Figure 18 sensors-23-01160-f018:**
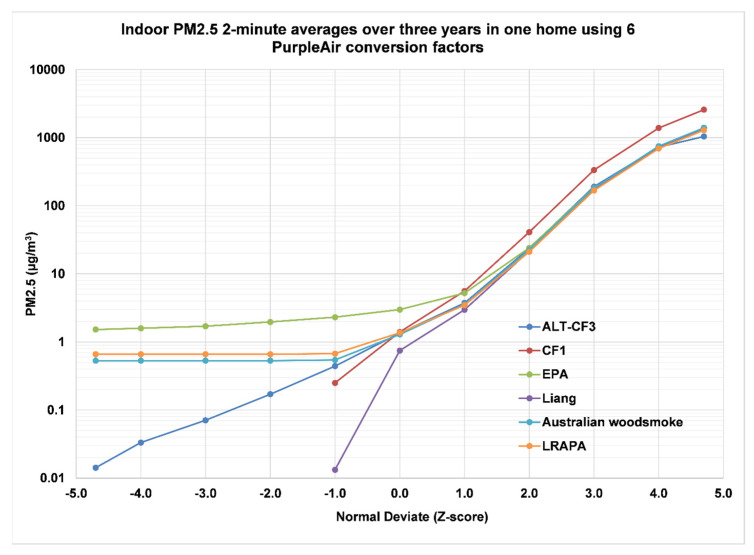
Comparison of 6 algorithms for determining PM_2.5_ estimates.

**Table 1 sensors-23-01160-t001:** PM_2.5_ (µg/m^3^) and associated precision at the Menlo Park site using the ALT-CF3 and Plantower CF1 algorithms.

	N Obs	Mean	Std.Err.	Lower Quartile	Median	Upper Quartile	Max
ALT-CF3							
Mean outdoor PM_2.5_	372,621	4.4	9.2 × 10^−3^	1.4	2.7	5.1	311
Indoor PM_2.5_	369,610	2.2	0.015	0.65	1.3	2.4	1050
Outdoor-infiltrated	372,621	0.78	1.6 × 10^−3^	0.25	0.5	0.9	55
Indoor-generated	369,610	1.78	0.015	0.21	0.6	1.5	1049
Precision outdoor	372,621	0.06	7.3 × 10^−5^	0.02	0.1	0.1	0.2
Plantower CF1							
Mean outdoor PM_2.5_	304,345	8.3	0.019	2.5	5.1	9.6	552
Indoor PM_2.5_	301,625	3.3	0.031	0.61	1.8	3.8	1874
Precision outdoor	304,345	0.07	9.1 × 10^−5^	0.03	0.1	0.1	0.2

**Table 2 sensors-23-01160-t002:** All PM_2.5_ data from Monitor 1 at the Oakmont site between 23 July 2019 and 28 September 2022.

	Valid N	Mean	Std. Err.	Min	Lower Quartile	Median	Upper Quartile	Max
MEAN 1 CF1	593,793	4.41	0.03	0	0.105	1.1	3.28	1678
MEAN 1 CF ATM	593,793	3.9	0.02	0	0.105	1.1	3.28	1118
MEAN 1 ALT-CF3	593,793	2.97	0.02	0.0063	0.543	1.167	2.39	865
precision 1 CF1	488,117 *	0.22	0.0004	0	0.04	0.098	0.25	1
precision 1 CF ATM	488,117 *	0.22	0.0004	0	0.039	0.098	0.25	1
precision 1 ALT-CF3	593,695	0.08	0.0001	3.02 × 10^−7^	0.029	0.062	0.11	0.87

* Missing 105,676 observations assigned a value of zero by the Plantower CF1 or CF_ATM algorithms.

**Table 3 sensors-23-01160-t003:** Number of zeros reported by CF1 (and CF_ATM) and percentage of total measurements.

Monitor/Sensor	Location	Zeros	Total	%
1a	Indoors	127,195	593,804	21
1b	Indoors	129,114	593,804	22
4a	Indoors	114,186	593,688	19
4b	Indoors	102,616	593,688	17
2a	Some outdoors	52,000	587,309	9
2b	Some outdoors	121,399	587,309	21
3a	Mostly outdoors	40,682	593,813	7
3b	Mostly outdoors	33,836	593,813	6

**Table 4 sensors-23-01160-t004:** LODs and percentage of observations exceeding the LOD for Oakmont comparing the ALT-CF3 algorithm to the Plantower CF1 algorithm.

Monitor	Location	Valid N	ALT-CF3 LOD (µg/m^3^)	ALT-CF3 N > LOD	ALT-CF3 (% > LOD)	CF1 LOD (µg/m^3^)	CF1 N > LOD	CF1 (% > LOD)
1	Indoors	406,108	0.99	233,900	58	2.9	177,908	44
2	Outdoors	253,454	0.92	203,384	80	9.9	39,487	16
2	Indoors	146,229	0.72	110,674	76	3.2	44,289	30
3	Outdoors	363,797	0.6	334,973	92	4.4	156,850	43
3	Indoors	42,304	0.52	42,114	99.6	2.9	32,150	76
4	Indoors	406,092	1.32	215,872	53	5.3	79,371	20

**Table 5 sensors-23-01160-t005:** All indoor PM_2.5_ data (µg/m^3^) and associated precision for monitors 1 and 2 compared for the ALT-CF3 and Plantower CF1 algorithms for the 19-month period 7/3/19 to 2/6/22.

	N obs.	Mean	Std.Err.	Lower Quartile	Median	Upper Quartile	Max
ALT-CF3							
Mean 1	662,773	4.2	0.020	0.93	1.8	3.3	735
Mean 2	662,635	4.0	0.017	0.92	1.8	3.4	539
precision 1	662,719	0.09	9.2 × 10^−5^	0.032	0.069	0.12	0.98
precision 2	662,561	0.10	1.2 × 10^−4^	0.043	0.080	0.13	0.99
Plantower CF1							
Mean 1	662,773	6.3	0.033	0.74	2.3	5.1	1149
Mean 2	662,633	6.1	0.029	0.70	2.3	5.1	883
precision 1	626,667	0.17	3.1 × 10^−4^	0.036	0.083	0.18	1
precision 2	617,016	0.21	3.2 × 10^−4^	0.066	0.12	0.22	1

**Table 6 sensors-23-01160-t006:** Calculations of the LODs (µg/m^3^) for 5 of 6 sites.

Site	Location	ALT-CF3	CF1/CF_ATM
Menlo Park	Outdoors	1.10	4.15
Oakmont	Indoors	0.99	2.90
	Outdoors	0.92	9.90
	Indoors	0.72	3.20
	Outdoors	0.60	4.40
	Indoors	0.52	2.90
	Indoors	1.32	5.30
Redwood City	Indoors	1.40	4.75
Alexander	Outdoors	0.97	2.69
Outer Sunset	Outdoors	1.12	3.27
Outer Sunset	Indoors	1.83	3.18
Statistics			
	Mean	1.04	4.24
	SD	0.38	2.06
	RSD	0.36	0.49

**Table 7 sensors-23-01160-t007:** Infiltration factors (F_inf_) and mean PM_2.5_ for outdoor-infiltrated (Out_inf_) and indoor-generated (In_gen_) particles, including the percent contribution to total indoor PM_2.5_ from the indoor-generated particles (In_gen_ (%)). These variables are shown for the entire datasets and for the subsets (1 and 2) created when the Forbidden Zone was violated by too many observations.

Outdoor Site	Oakmont	Bennett Valley	Redwood City	Alexander Ave	Menlo Park	Outer Sunset
Indoor site	same	Oakmont	same	same	same	same
Finf	0.34	0.38	0.36	0.17	0.085	0.33
PM_2.5_ Outinf	3.2	2.28	3.0	0.99	0.31	2.6
PM_2.5_ Ingen	1.7	1.3	2.6	0.43	1.8	10
Ingen (%)	34	36	46	31	85	80
Finf 1	0.34	0.36	0.23	0.080	NA	NA
PM_2.5_ Outinf 1	2.2	4.10	2.9	0.44	NA	NA
PM_2.5_ Ingen 1	2.1	2.2	2.3	0.51	NA	NA
Ingen 1 (%)	48	35	45	54	NA	NA
Finf 2	0.14	0.22	NA	NA	NA	NA
PM_2.5_ Outinf 2	0.38	0.75	NA	NA	NA	NA
PM_2.5_ Ingen 2	1.5	1.1	NA	NA	NA	NA
Ingen 2 (%)	80	59	NA	NA	NA	NA

Outinf = outdoor-infiltrated; Ingen = indoor-generated; 1, 2 = subsets 1, 2; NA = Not Applicable.

## Data Availability

All raw data publicly available from PurpleAir websites (https://www2.purpleair.com/, accessed on 1 January 2023), (https://api.purpleair.com/, accessed on 1 January 2023). All RCS regression analyses are available on request from the corresponding author.

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
