# Peer review of "Long-Term Indoor-Outdoor PM2.5 Measurements Using PurpleAir Sensors: An Improved Method of Calculating Indoor-Generated and Outdoor-Infiltrated Contributions to Potential Indoor Exposure"

_sensors, 2023, doi:10.3390/s23031160_

Round 1

Reviewer 1 Report

As it said, this paper develops a  method of identifying periods when the infiltration factor is not constant, and searching for periods when it is relatively constant. It is important in our real life to evaluate the PM2.5. This paper has also presented the model results with many details. However,

1.this paper said the Long-term indoor-outdoor PM2.5 measurements using PurpleAir sensors, so what is the sensors? How is the sensor hardware? Why did you use the sensors? The more details about the sensors are lack.

2. The data analysis and results are more suit for the energy/environment journals.

Author Response

1.this paper said the Long-term indoor-outdoor PM2.5 measurements using PurpleAir sensors, so what is the sensors? How is the sensor hardware? Why did you use the sensors? The more details about the sensors are lack.

Thank you for identifying this oversight.  The PurpleAir monitors are described in a new section 2,1 under Materials and Methods. We have also added a picture of a monitor installed in an outdoor location to the Supplementary Information.

  1. The data analysis and results are more suit for the energy/environment journals.

We do agree with the reviewer that these findings are very suitable for the environmental journals.  But the entire dataset that we use is provided by the low-cost environmental sensors, so we think that the paper also fits well in the Sensors journal.  Also, the approach of determining the calibration factor for the sensors based on developing an algorithm to replace the algorithms offered by the sensor manufacturer is squarely within the purview of the Sensors journal. Finally, the LOD discussions refer only to low-cost sensors

Reviewer 2 Report

The paper presents relevant findings and detailed analysis; however, excessive self-citation is a major concern. 

Author Response

The paper presents relevant findings and detailed analysis; however, excessive self-citation is a major concern. 

We agree that we have cited many of own papers (about 30 out of 50 total).  However, we have both spent our entire careers (now reaching 45 years each) in this precise area of human exposure. We believe we have cited many of the references that are among the leading references in this area, such as the Harvard 6-City Study.  We think that having lots of references adds to the paper if it awakens a desire in the reader to go beyond our findings.  That said, we have added 5 more references (51-54 and 56) in the section on studies looking at the Limit of Detection, and we are happy to note that none of the five included us as authors.  We would gladly add references that the reviewer believes would augment the paper.

Reviewer 3 Report

Using data recorded by PurpleAir sensors at six locations in California, US, Wallce and Ott evaluated the performance of the ALT-CF3 algorithm for calculating PM2.5 mass concentrations and estimated the limit of detection (LOD) of the sensors. The long-term indoor-outdoor PM2.5 relationships were explored using the regression method. This work is within the scope of Sensors and of interest to the air quality research community.  However, major revisions are needed before it is accepted for publication.

Major comments:

1.      The Introduction needs improvement. The authors actually focused on three parts, i.e., the evaluation of the performance of the ALT-CF3 algorithm, the calculation of the LOD, and the estimate of the outdoor/indoor contributions to the indoor PM2.5. However, only the third part was mentioned in the Introduction. While some background information needs to be added for the first two parts, descriptions on part three should be shortened.

2.      Site descriptions should be moved to Section 2. A summary of all the sites are preferred.

3.      LOD calculations for several locations are missing.

4.      The treatment for data with I/O ratios >1 is inconsistent.  L355 states data having I/O ratios > 1 were dropped, but in Figures 3, 5, 6, 11, 12, 14 -17, such data were used for regression, which would influence the results, and thus the associated conclusions would probably be different.

Minor comments:

1.      When describing data points in the Forbidden Zone, the authors used “a large number of values” several times, which is rather arbitrary. Please be more specific.

2.      It is hard to read the dates from the daily I/O ratio scatter plots. Also, what does the red line represent? And, these plots should be moved to the Supplement.

3.      L172: replace “poor” with “low”.

4.      L178: What is the average diameter used for each size fraction?

5.      L198 and L251: The reference should be [48].

Author Response

  1. The Introduction needs improvement. The authors actually focused on three parts, i.e., the evaluation of the performance of the ALT-CF3 algorithm, the calculation of the LOD, and the estimate of the outdoor/indoor contributions to the indoor PM2.5. However, only the third part was mentioned in the Introduction. While some background information needs to be added for the first two parts, descriptions on part three should be shortened.

Thank you for this comment.  We agree, and have added paragraphs on the two parts of our study that were missing from the Introduction.  These paragraphs are general but they point to the full descriptions provided in the Materials and Methods section (2.2 and 2.3).

Regarding the performance of the ALT-CF3 algorithm, we have added paragraphs in the Discussion section together with a new Figure 18 comparing its performance with 5 other models..

  1. Site descriptions should be moved to Section 2. A summary of all the sites are preferred.

We have moved all site descriptions to Section 2 (2.1.1 to 2.1.6) The six sites appear together in Tables 6 & 7 so we think a table here, although desirable perhaps, would be duplicative.3LOD calculations for several locations are missing.

  1. 3. LOD calculations for several locations are missing.

This is an important comment.  We have added six more LOD calculations for indoor and outdoor locations at three additional sites, so we now have completed 5 of the 6 sites.  The number of LOD measurements is best presented in a Table 6, which we have added to the Discussion Section.  We also added in that section a short survey of other LOD estimates from field and laboratory investigations. Finally, we added a new Figure 16 showing how the revised approach to include 1000 contiguous concentrations rather than 100 has reduced the noise and produced a relatively smooth function allowing easier identification of the LOD.  

  1. The treatment for data with I/O ratios >1 is inconsistent.  L355 states data having I/O ratios > 1 were dropped, but in Figures 3, 5, 6, 11, 12, 14 -17, such data were used for regression, which would influence the results, and thus the associated conclusions would probably be different.

This is a very good catch by the reviewer.  The graph referred to is Figure 7 and was the first graph that we prepared.  Of course, many later graphs had multiple I/O ratios >1.  We reconsidered dropping values >1 at that point, concluding that it is rather arbitrary (why not choose 2, say, or 0.8?) so we stopped the practice of dropping such values. We should have gone back to Figure 7 at that point and revised it. Mea culpa. We have restored the 4 points that were dropped from Figure 7 and have created a revised version of Figure 7 with 411 total observations.  All 4 points occurred in the April-October time frame, for which we presented no graph because of too many points violating the Forbidden Zone boundary.  This general conclusion did not change on adding the 4 points, although the slope increased by a small amount..  Figure 8 with the November-March time frame included none of the 4 points and therefore required no change. The reviewer is correct that dropping or retaining these values will give different slopes. and different intercepts. However, the arbitrary nature of choosing a cutoff value opens up a near-infinity of different results. Therefore we have chosen to accept all measured values.

Minor comments:

  1. When describing data points in the Forbidden Zone, the authors used “a large number of values” several times, which is rather arbitrary. Please be more specific.

This is a good point. I replaced these 4 cases with the exact number and the percentage of values falling in the Forbidden zone.  The percentages ranged from15-33%. This raises the question of how many observations in the Forbidden Zone are “too many”. It would seem that 1% would be acceptable, and 15% might not.  Perhaps 5% or 10% would be reasonable values. 

  1. It is hard to read the dates from the daily I/O ratio scatter plots.            Using the View/Page Width option allows the dates to be quite readable.
  2.  Also, what does the red line represent? And, these plots should be moved to the Supplement.
  3. The red line is a distance-weighted regression (LOESS) fit. This information has been added to each figure caption.   We think at least one plot should remain in the paper because it will be useful to the reader for understanding our choices of how to find subsets of the data that might have near-constant infiltration plots.  We have moved two of the four plots to a Supplement.
  4. L172: replace “poor” with “low”.  Done.
  5. L178: What is the average diameter used for each size fraction?

The geometric mean of the two endpoints of the size category. This is used by several manufacturers of particle monitors, such as TSI.  A choice of the arithmetic mean results in somewhat higher PM2.5 estimates (~12%). We have added a sentence referring to the geometric mean

  1. L198 and L251: The reference should be [48].

Actually, the method was first published in ref [49], but it is true that ref [48} also applies the method.  

Reviewer 4 Report

the article presents the novelty of low cost detectors, which could be very interesting to be able to easily measure particles in different environments. the information presented is interesting and very useful.

Author Response

Thank  you for the supportive comments

Round 2

Reviewer 1 Report

The paper was revised well.

Reviewer 2 Report

Can be accepted 

Reviewer 3 Report

Agree for publication.